# A Comprehensive Survey on Nanophotonic Neural Networks: Architectures, Training Methods, Optimization, and Activations Functions

**DOI:** 10.3390/s22030720

**Published:** 2022-01-18

**Authors:** Konstantinos Demertzis, Georgios D. Papadopoulos, Lazaros Iliadis, Lykourgos Magafas

**Affiliations:** 1Department of Physics, Faculty of Sciences, Kavala Campus, International Hellenic University, St. Loukas, 654 04 Kavala, Greece; gebapad@teiemt.gr (G.D.P.); lmagafas@otenet.gr (L.M.); 2School of Science & Technology, Informatics Studies, Hellenic Open University, 263 35 Patra, Greece; 3School of Civil Engineering, Faculty of Mathematics Programming and General Courses, Democritus University of Thrace, Kimmeria, 691 00 Xanthi, Greece; liliadis@civil.duth.gr

**Keywords:** nanophotonic neural networks, photonic neural networks, optical neural networks, optical interference unit, optical non-linear unit, optical activation function

## Abstract

In the last years, materializations of neuromorphic circuits based on nanophotonic arrangements have been proposed, which contain complete optical circuits, laser, photodetectors, photonic crystals, optical fibers, flat waveguides and other passive optical elements of nanostructured materials, which eliminate the time of simultaneous processing of big groups of data, taking advantage of the quantum perspective, and thus highly increasing the potentials of contemporary intelligent computational systems. This article is an effort to record and study the research that has been conducted concerning the methods of development and materialization of neuromorphic circuits of neural networks of nanophotonic arrangements. In particular, an investigative study of the methods of developing nanophotonic neuromorphic processors, their originality in neuronic architectural structure, their training methods and their optimization was realized along with the study of special issues such as optical activation functions and cost functions. The main contribution of this research work is that it is the first time in the literature that the most well-known architectures, training methods, optimization and activations functions of the nanophotonic networks are presented in a single paper. This study also includes an extensive detailed meta-review analysis of the advantages and disadvantages of nanophotonic networks.

## 1. Introduction

Artificial intelligence (AI) [1] enables machines to be trained so as to perform particular tasks, learn from experience, adapt to or interact with the environment and perform realistic anthropomorphic tasks [2]. Contemporary AI is one of the fastest evolving fields of information technology, in which high-level algorithmic approaches and tools descending from applied math’s and engineering are used [3,4,5]. Most AI applications—from computers playing chess to automatically driven cars—are based to a great extent on the intelligent technologies of neural networks (NNs) [6] for the processing of multidimensional big data, with a view to revealing the hidden knowledge that is included in these groups [7,8].

In classic von Neumann architecture, where the computations are restrained by the speed of the channel between computation and memory (also known as the von Neumann congestion), even the important innovations on problems such as the shrinking of complete circuits, the reduction in their power needs and the decrease of temperature emitted by them cannot achieve the anticipated increases in their computing power [9,10]. Even with the introduction of a graphics processing unit (GPU) as an extra processor for the improvement of graphic interface and the performance of tasks of high-level processing, or the introduction of Google’s tensor processing unit (TPU) as the most powerful adapted processor for the performance of AI procedures, the capabilities of traditional systems seem unable to cope with the demands of modern technology and the uninterrupted flow of the data produced, even if they have offered some of the most important innovations [10,11,12].

The biggest challenge in this field is the development of fully functional and utilizable neuromorphic systems-on-chip (NSoC) [10,13], which will be able to approach the biological human intelligence, performing the same tasks that the human brain effortlessly performs in no time at all and without remarkable consumption of resources and energy [14]. The neuromorphic computation comprises the creation of neural networks in matter, where the neurons of a physical device are connected with the corresponding synapses of physical devices [15]. The main motive for the neuromorphic computation is the time needed to process the computations and the energy performance provided by a distributed architecture, which avoids the energy turbulence of data between the memory and the CPU [16].

The NSoCs based solidly on previous computational technology overcome the von Neumann congestion, massively use simultaneous computational procedures and are tolerant to faults [17]. Essentially, they form the way in which neural networks function, conveying information in the same temporal and spatial way as the human brain. Furthermore, taking advantage of techniques such as the memristors [10], they are capable of modeling learning skills; that is, the adjustment ability of synapses in storing and conveying information depending on the evolution of a dynamic situation [16].

However, the most important development in the application efforts and standardization of NSoCs is spotted in the expanded efforts for developing fully optical neural networks (ONNs), also known as photonic neural networks (PNNs) or nanophotonic neural networks (NNNs) [15,18,19,20,21]. The previously mentioned systems are based on the evolutions of optical technology and the most recent research concerning photonics. Photonics is the science and technology field that deals with the creation, control and detection of photons, especially in the area of visible light and the near-infrared electromagnetic spectrum (wavelength, polarization, transmission rate, etc.) and the great potentials of their interconnection [20,22,23]. It is directly related, in basic as well as applied research, to quantum optics as to how linear transformations can be applied with the minimum energy consumption and with the slightest latency time on neuron level and with optoelectronics in the study of active and passive materials that interact electrically with light [24,25].

Many efforts have been made in recent years to shift from conventional electronics to optical circuits. This review records the most recent research to clarify how close we are to the complete transition to photonic arrangements and their exceptional prospects. Moreover, the main contribution of this research work is that it is the first time in the literature that the most well-known architectures, training methods, optimization, and activations functions of the nanophotonic networks are presented in a single paper. Additionally, the manuscript includes an extensive detailed meta-review analysis of the advantages and disadvantages of nanophotonic networks.

The rest of the paper is organized as follows: Section 2 contains the principles of light and matter interaction. Section 3 describes the current state of research in neuromorphic processors using photon circuits and Section 4 analyzes these architectures. Section 5 is dedicated to the training operation of PNNs. Section 6 summarizes the most common activation functions that are used. In conclusion, Section 7 presents the final remarks and perspectives.

## 2. Nature of Light

In general, the interaction of light with matter and its diffusion inside it, is sufficiently described by optics (geometric and wave optics) [22,26]. While studying various optic phenomena when the intensity of impacting radiations is small, the response in general and all the individual optic properties/parameters of materials (e.g., refraction index, absorption factor, polarization, etc.) remain stable and independent of intensity [27]. However, when the intensity of radiation is high, as it happens with laser, for example, and in particular with a focused-beam laser of great power, it has been proved experimentally that the optic response of matter and the optical parameters are modified, often significantly, and become dependent on the intensity of the radiation [28]. It is then that various extremely interesting phenomena take place, which are not detected in the case of low intensity impacting radiation [29]. These phenomena cannot be interpreted considering the linear response of matter, as it is expressed in the fundamental linear relation between the cause, i.e., the *E* electric field, and the result, i.e., the *P* inductive polarization [22,30,31]:(1)$$\vec{P}=\varepsilon_{0}\chi^{\left(1\right)}\vec{E}$$
where $\chi^{\left(1\right)}$ is the linear susceptibility and $\varepsilon_{0}$, *ε* is the intra-electrical invariant of vacuum and matter, respectively. For the susceptibility $\chi^{\left(1\right)}$ and the refraction index *n*, it is true that [24,28]:(2)$$n^{2}=\frac{\varepsilon }{\varepsilon_{0}}=1+\chi^{\left(1\right)}$$

In the case of high-intensity radiation, conditions of high rate appear in the expression of polarization, the contributions of which are essential and cannot be omitted. These phenomena are the result of modification of the optic properties of matter due to the powerful electric field, and the non-linearity of these phenomena is attributed to the fact that the response of matter is a non-linear function of the intensity of radiation [32].

One of the most important consequences of the linear response of matter under the influence of intense fields, apart from the alterations of the properties of matter, is that if there are different beams going through the same region of a non-linear medium simultaneously, they can interact with one another through matter [30]. Going this reasoning one-step further and taking into consideration the principle of superposition, according to which a beam can be considered as the superposition of two beams of the same polarization, frequency and direction, we can assume that a beam can interact with itself [25,26,28].

As far as the procedure that causes the appearance of a non-linear optic behavior is concerned, when some radiation impacts on some material, it causes changes in the spatial and temporal distribution of the electric charge, inducing electric dipoles, the macroscopic result of which is the creation or modification of the polarization of the material [33]. For low values of the *E* field, the *P* polarization is analogous to the *E* field that caused it and the elementary dipoles, when oscillated, emit radiation of the same properties as the impacting radiation. Nevertheless, for high intensities of the *E* field, the radiation emitted by the elementary dipoles is not in correspondence with the electric *E* field that caused it. This can be explained by the fact that the captivated electrons of atoms/ions/unitary cells of crystal (or the structural unit in general) are forced to great displacements from their balance position. As a result, the motion of electrons cannot be described by the model of harmonic oscillator of Lorentz. Then, the radiation emitted contains frequencies different from those of the initial stimulating radiation. This practically means that it is possible to modify the impacting radiation itself with the addition of new frequencies, for example. In this way, non-linear phenomena can be interpreted and applied, which can explain how a beam of light can interact with one another (or with itself) creating amplification of light through light, merging of a beam with another one, production of new frequencies, etc. [34].

Based on what was previously mentioned, for high intensity of the electric field (e.g., for *Ε* > 10^5^ V/cm) where the presence of non-linear phenomena becomes significant, in the equation describing the polarization conditions of a higher rate appear, and the polarization is presented as an expansion of the Taylor sequence according to the following form [33,35]:(3)$$\vec{P}\left(t\right)=\varepsilon_{0}\left[\chi^{\left(1\right)}\vec{E}\left(t\right)+\chi^{\left(2\right)}{\vec{E}}^{2}\left(t\right)+\chi^{\left(3\right)}{\vec{E}}^{3}\left(t\right)+\cdots \right]$$
where $\chi^{\left(2\right)}$ is the second-rate susceptibility, $\chi^{\left(3\right)}$ is the third-rate susceptibility and so on and so forth. The susceptibilities are generally tensors, so, for instance, the first-rate susceptibility $\chi^{\left(1\right)}$ is a second-rate tensor with 3 × 3 = 9 elements and the corresponding polarization is presented by the following form [25,34,35]:(4)$$\left[\begin{array}{c} P_{x}^{\left(1\right)}\\ P_{y}^{\left(1\right)}\\ P_{z}^{\left(1\right)} \end{array}\right]=\varepsilon_{0}\left[\begin{array}{ccc} \chi_{xx}^{\left(1\right)} & \chi_{xy}^{\left(1\right)} & \chi_{xz}^{\left(1\right)}\\ \chi_{yx}^{\left(1\right)} & \chi_{yy}^{\left(1\right)} & \chi_{yz}^{\left(1\right)}\\ \chi_{zx}^{\left(1\right)} & \chi_{zy}^{\left(1\right)} & \chi_{zz}^{\left(1\right)} \end{array}\right]\left[\begin{array}{c} E_{x}\\ E_{y}\\ E_{z} \end{array}\right]\;\mathrm{or}\;P_{i}^{\left(1\right)}=\varepsilon_{0}\;\sum_{j}\;\chi_{ij}^{\left(1\right)}E_{j}$$
where i,j=x,y,z.

Similarly, the non-linear second-rate susceptibility $\chi^{\left(2\right)}$ is a third-rate tensor $\chi_{ijk}^{\left(2\right)}$, whereas the third-rate susceptibility $\chi^{\left(3\right)}$ is a fourth-rate tensor $\chi_{ijkl}^{\left(3\right)}$. In the case that the medium displays losses, the susceptibility $\chi^{\left(1\right)}$ is a complex quantity with its real part being connected to the linear refraction index *n* and its imaginary part being connected to the factor of linear absorption through the following relations [21,22,26]:(5)$$\chi^{\left(1\right)}=Re\left(\chi^{\left(1\right)}\right)+i\left[Im\left(\chi^{\left(1\right)}\right)\right]$$
where $Re\left(\chi^{\left(1\right)}\right)\propto n_{0}\;\mathrm{and}Im\left(\chi^{\left(1\right)}\right)\propto \alpha_{0}.$

The equivalent relations also apply to the non-linear high-rate susceptibilities, which are also complex numbers with real and imaginary parts equivalent to the corresponding refraction indexes and absorption factors, which correspond to the non-linear refraction index and the non-linear absorption factor. When an intense laser beam passes through a material, the electric field of the beam can induce a change in the refractive index of the material that is proportional to the intensity of the beam. This non-linear effect is called the Kerr effect. The total refractive index of the material is the sum of the refractive index, *n*_0_, with no laser beam present and the term *n_2_ I*, where *n*_2_ is the second-order non-linear refractive index and *I* is the intensity of the beam. The change in refractive index can be positive or negative.

It is also important to point out that the calculation of an observed value in a system of photonic arrangements disrupts the system and, therefore, it shifts to a quantum condition in which the repetition of calculations of the same property leads to the same result. Thus, the following quantum conditions are observed [21,24,36]:(1)|𝐸⟩: Quantum condition where, if power is calculated, the result will be *E*.(2)|𝑝⟩: Quantum condition where, if momentum is calculated, the result will be *p*.(3)|𝑥⟩: Quantum condition where, if position is calculated, the result will be *x*.

In a general condition |𝜓⟩, the possibilities of calculating various physical properties are uncertain, that is there is the possibility 𝑃_1_(𝐸) of calculating the value of energy as *E*, 𝑃_2_ (𝑝) is the possibility of calculating the value of momentum as *p* and so on and so forth. In a |𝜓⟩, condition system, after the calculation, for example, of energy with an *Ε*_1_, result, the wave function is disrupted and collapses (transforms) into a new condition |𝛦_1_⟩, so that the repetition of the same calculation gives the same result. Respectively, in a |𝜓⟩, condition system, after calculating for example the momentum with the result *p*_1_, the wave function is disrupted and collapses (transforms) into a new condition |𝑝_1_⟩, so that the repetition of the same calculation gives the same result [37,38].

The conditions |𝑥⟩ and |𝑝⟩ cannot coincide, because the calculation of position (e.g., with photon scattering of short wavelength) alters the momentum. Consequently, there is no certainty about the momentum and the position of a particle as the values observed are random variables, in the sense that for every value of the spectrum of an observed physical quantity, a quantum width of probability for the calculation of this particular value corresponds to it. The total amount of quantum widths of probability for a spectrum of an observed physical quantity fully determines the quantum condition of the system. In that sense, one of the targets of photonic systems is the calculation of these widths of probability with the use of results of analytic methods [39,40].

In conclusion, taking advantage of the binary nature of light and all the other characteristics that render it the fastest means of communication, the verge of modern investigation is focused on developing photonic neuromorphic processors [38,41].

## 3. Photonic Neuromorphic Processors

The investigation into the development of photonic neuromorphic processors with passive optic circuits, focuses on advantages such as the following [21,22,24]:(1)Significant reduction of energy consumption in the applications of logical circuits as well as in data transfer.(2)Exceptionally high operating speeds with no energy consumption other than on the transmitters and the receivers.(3)Distribution of the computing power in the whole network, with each neuron performing simultaneously small parts of the whole computational activity.

Nevertheless, a big obstacle of photonic circuits has been the great volume of optical devices and the absence of susceptibility in contrast to the traditional integrated electronic circuits [13,18,21]. The materializations mainly on silicon (Si) and additionally on indium phosphide (InP) constitute a great innovative breakthrough in the materialization of integrated photonics, which is a reality nowadays. Especially, the materializations of integrated photonics with silicon as the construction material have proved to be excellent as they are transparent for wavelengths of 1270 to 1625 nm that are used in communications, and the refraction index with a breadth of 3.48 in 1550 nm guarantees great resistance, while at the same time it can be checked thermically, electrically, mechanically or chemically [16,17,32].

Taking advantage of the properties above, silicon has been widely used for the materializations of passive elements, like waveguides, modulators, splitters, couplers and filters. On the other hand, indium phosphide allows for the materialization of monolithically integrated solutions, which include a combination of passive and active devices, such as lasers and amplifiers. Moreover, the integrated photonic technology offers the prospect of reducing the order of magnitude of the integration into nano-levels with all the significant advantages that the aforementioned reduction of size brings about, such as the reduction of energy footprint, smaller size, etc. [40,42].

Therefore, a multitude of arrangements of optical spare parts and integrated photonic circuits is already available, which results in the appearance of significant progress in the field of photonic neuromorphic, with the development of nanophotonic neural networks with either applications or waveguides or free-space optic [20,36,43].

## 4. Architectures

The philosophy behind the use of photonic circuits [22,44,45] in nano-arrangements [46,47,48] and materials is based on the need for significant improvement of the speed of transmitting and processing data and for an improvement of the energy efficiency of devices. The PNN materializations based on the aforementioned materials, which are present to this day, are classified into two main categories: with memory (stateful) and without memory (stateless), as they are concisely presented in Figure 1 [18,21,49]:

Moreover, the PNNs are classified according to design (integrated or free-space optic) and according to optical training ability (trainable) or reference only (inference). The types of networks that have been applied in modern neuromorphic technology and their respective modifications are thoroughly explained below [18,49].

### 4.1. Perceptron

The perceptron model consists of a single neuron, is the simplest autonomous system in existence and performs a particular task. This unique neuron of the system has a particular number of connections deriving from other neurons. The perceptron’s development into OΝNs is the most fundamental scientific field, with many articles having been published with respective materializations [20,50,51,52,53]. An all-optical neural network (AONN) architecture with a hidden layer is presented in Figure 2 [54].

It is based on free-space optics, without the use of light wave guidance and integrated circuits, encoding the input signals with alterations in the illuminating power. During its linear operation, the light impacts on different areas of the surface of a spatial light modulator (SLM) representing the knots $v_{i}$ of the input layer of a ΝN. With a special grid coating, the impacting light beam can be split in different *j* directions with weight *W_ij_*. The SLM is placed in the rear focal layer of the lenses, which apply Fourier transformation and sum up all the diffracted beams on the focal point as follows [49,51,54]:(6)$$z_{i}=\sum_{j}\;W_{ij}v_{j}$$


This is as it happens with every knot of a conventional ΝN.

The non-linear operation is accomplished through electromagnetically induced transparency (EIT), which is based on quantum phenomena and is produced by laser-cooled atoms ^85^Rb, in a magneto-optical trap (MOT) [55,56]. The materialization of this particular architecture is shown in Figure 3 [40,42,54,57,58]:

The light beam from the laser single-mode fiber (SMF), which constitutes the encoded input layer, is aligned (L1) and impacts on the first modulator (SLM1), which, in turn, emits four different beams. These are directed towards the L3 lens as $v_{j}$ inputs, while at the same time the C1 camera, through a special flip mirror (FM), records and calculates their values. Through the L4 and L5 lens system, the non-linearity is introduced by the MOT and afterwards the beams are directed towards the SLM2, after being recorded by the C2 camera first. Finally, the next layer (output layer), which consists of the SLM2 and the L7, L8 and L9 lenses, transforms the four beams into two, which are recorded by the C3 camera [27,29,54]. It is important to mention that the single-layer perceptron’s optically implement matrix multiplications. Implementation of matrix multiplication in the optical domain has been a topic of research for decades, and has been shown in free space through the use of beam splitters or Mach–Zehnder interferometers as well as in integrated photonic circuits through the same mechanisms, for application in optical signal processing and reconfigurable optical neural networks. Recently, diffractive neural network architectures have been proposed, in which these matrix multiplications are performed by diffractive elements. This marked the beginning of optical data processing through diffractive neural network inference, although the fabrication methods applied are only suitable for devices operating with a low neuron density.

For the evaluation of this architecture, a classification of the different stages of an Ising model [59] has been carried out, giving similar results compared to a ΝN created by a computer as these are represented in Figure 4 [37,54].

It is obvious that this particular materialization can fully substitute a ΝN created by computers, with the only exception being the fact that commercial SLMs are not fast enough compared to a modern computer [39,54,60].

### 4.2. Multilayer Perceptrons

A modified model of multiple sensors is called multilayer perceptron, in which between the input and output layers intervene in one or more hidden layers. The data flow in such a network is always from its inputs to its outputs and there is no feedback loop. We also assume that the neurons in every layer interact only with those neurons that belong to their directly adjacent layers. In other words, the first hidden layer accepts the values of the input layer, and the results of the first hidden layer go through the second hidden layer, whose results then go through to the third layer until they finally reach the final output layer. The materialization of the nanophotonic [46,48] multilayer perceptron of Figure 5 is based on the use of nanophotonic circuits that process coherent light [49,61,62,63,64].

As shown in Figure 5, the basic theoretic block of ΝN with the hidden layers (grey) (a), is transferred to optical level operation (b), using two basic optical parts (c), and in particular, an optical interference unit (OIU), which performs multiplication of matrixes, and an optical non-linear unit (ONU), which materializes the activation function. All the above are given briefly and in an integrated circuit form (d). The OIU consists of ranks of special programmed Mach–Zehnder interferometer (MZI). The MZIs convert the phase differences of light into amplitude differences (modulation). The structure of a MZI is shown in Figure 6 [18,51,52,61,64].

Finally, the modification of input *M* matrix of every *i^th^* rank into a matrix product is accomplished as shown in Equation (7) according to the singular-value decomposition (SVD) [18,49,64,65]:(7)$$M^{\left(i\right)}=U^{\left(i\right)}\Sigma^{\left(i\right)}V^{*\left(i\right)}$$
where U is a m×m real or complex unitary matrix, Σ is a m×n orthogonal diagonal matrix with no negative values in the diagonal and $V^{*}$ is the conjugate transpose of *V*, which is a n×n real or complex unitary matrix. It must be highlighted that through the *M* matrix, the weight matrix *W_i_* of the ΝN is transferred to the optical circuit [22,66,67].

The unit of the transfer function in this particular research paper has not yet been experimentally materialized and can only be simulated in a computer, with transformation of the signals from the optical to the electrical layer of operation and vice versa.

The experimental study and the training of OΝN was put into practice on a computer first, initially through the application of voice recognition with 76.7% accuracy. Then, the already familiar diagnostic tool of digital identification, the Modified National Institute of Standards and Technology database (MNIST), was used, in which accuracy of results reach 95%, with the highest known value being 99%. This last conclusion shows the potentials and dynamics of mechanical learning in this particular field [18,48,52,68].

### 4.3. Deep Photonic Neural Networks

Deep neural networks (DNNs) are used in solving complex problems of high complexity like medical image analysis, speech recognition, language translation, image classification and many more [52]. However, as the number of layers increases, its structure becomes more complex and this result in the input of a great computational load on the processor. Consequently, the training time increases and so does the energy consumption. These restrictions created the need for the materialization of PΝNs of many layers (deep photonic neural networks—DPNNs), since the advantages of photonic transmission speed and the minimum energy consumption are indisputable [47,49,52].

The construction of standardized, fully optical circuits, with many layers is a true challenge nowadays. An arrangement for the materialization of a DPNN is shown in Figure 7 [69].

In this particular architecture, the layers of ΝN are substituted with photonic grids, in which instead of nodes with neurons we have waveguides. The interconnection between layers is accomplished through coupling devices with weighted cross connectors so that the desired output from the network can be achieved. The coupling devices, which are responsible for the control of photons, consist of optical splitters and optical combiners for which the following relation is in effect:(8)$$c_{i}=\sum_{j}\;w_{ij}\cdot s_{j}$$

The weights $w_{ij}$ are controlled by external parameters until the network reaches the ideal output and be led to a condition of stable weights (training), where we obtain the following [18,19,22,40,42,46,49,52,69]:
(9)$$\alpha_{i}^{\left(l\right)}=\sum_{j=1}^{N_{\left(l-1\right)}}w_{ij}f_{j}^{\left(l-1\right)}\left(a^{\left(l-1\right)}\right)$$

### 4.4. Convolutional Neural Networks

The convolutional neural networks (CNNs) [70] adopt a different approach in their organization as they take advantage of the hierarchical standard of the input data, creating more complex, but fewer and simpler patterns, in their architecture. The nanoscale neuron size not only provides the advantage of a high neuron density, but also results in a short distance (the operative distance, i.e., the distance between the input and output planes, is one to three orders of magnitude smaller than that in other implementations) and more connections between the neurons due to the increased diffraction angles. These features lead to three orders of magnitude increase in the operational frequency, and thus in the operations per second (FLOPS) compared with the devices in the THz region. In this regard, smaller feature sizes can be achieved (<10 nm), potentially creating a completely new platform for smart systems based on CNN.

The architecture of a CNN is analogous to the one of the convertibility patterns of neurons of the human brain and was inspired by the organization of the optical cortex. More analytically, a CNN is a deep learning algorithm, which can take an image at the input, assign the appropriate weights to some of its various characteristics and, consequently, be able to differentiate one from the other. In other words, it has the ability to successfully record the spatial and temporary dependencies in an image through the application of relevant filters. Thus, a better adjustment to the total data is accomplished due to the decrease in the number of parameters that are involved and the reuse of weights [71]. In other words, the network can be trained to better comprehend the structure of an image for example, while the preprocessing that is needed in a CNN is smaller when compared to other classification algorithms. The outcome is that CNNs have an advantage over the ΝN with perceptron’s because the latter are prone to data overload due to the full connection of their knots.

There are several suggestions with CNNs that have been published such as [10,53,72]. A hybrid multilayer optical-electrical ΝN based on an optical matrix multiplier is presented in Figure 8 [73].

In every one of the network’s layers, the inputs $x^{\left(k\right)}$ are multiplied with the corresponding weights $W_{i}^{\left(k\right)}$, which are encoded as optical signals with homodyning between each pair of signal weight. The electronic signals that derive are then subject to a non-linear transition function *f* and are converted to serial signals. Then, they are converted once again to optical signals and are sent to the input of the next layer. This optical system can be used for fully connected as well as CNNs and allows for the inference of conclusions as well as the training in the same optical device.

Another suggestion of a CNN with full use of optical convolutional neural networks (OCNN) is presented in Figure 9 [18,49,70,74,75].

The architecture consists of layers separated in an OIU based on MZI, which performs linear operations on the center panel (convolutional and pooling), one part for the input of the non-linearity unit and a splitters network of 3 dB, for the reorganization of data that the CNN is processing (re-shuffling).

The separators are programmed to introduce the appropriate time lag so that, at the output of the network layers, the signals could synchronize in time and form a new data entry for the input into the matrix nucleus of the next layer. It can be calculated that with this particular architecture the processing would be 30 times faster than that of an especially purpose-built electronic processor for CNNs with the same power consumption. As a result, such a system could play a significant role in the processing of thousands of terabytes of image and video data that are produced every day on the internet [41,52].

### 4.5. Spiking Neural Networks

The *spiking neural networks* (SNNs) [76,77,78] are networks that imitate more than any other the biological ΝΝs. Apart from the neural and synaptic condition, the SNNs incorporate the concept of time in their operating model. The idea behind this is that the neurons in a SNN should not trigger and be triggered in every propagation circle, as in standard networks of multiple layers with perceptron’s. As it happens with the biological neurons, when the dynamics of their cell membrane reaches a particular value, which is called action potential, then the neuron triggers and produces a signal that travels to other neurons, which, in turn, increase or decrease the dynamics of their cell membrane according to this particular signal. The SNNs use peak sequences as mechanisms of internal information presentation, in contrast to the usual continuous variables, while at the same time having equal, if not better, performance in computational cost to the traditional NNs [79,80,81].

In the field of optical SNNs, many studies have been conducted in the past years [82,83], initially taking advantage of the fast optical elements used in the construction of big systems with optical fibers. Despite the significant advances to build active optical artificial neurons using for example phase-change materials, lasers, photodetectors and modulators, miniaturized integrated sources and detectors suited for few-photon spike-based operations and of interest for neuromorphic optical computing are still lacking. The successful applications finally led to the completion of arrangements, aiming for greater scalability, increase of energy efficiency, reduction of cost and flexibility in the environmental fluctuations.

In a survey, the use of a graphene laser is recommended as an artificial neuron, which is the fundamental element for the processing of information in the form of spikes. Moreover, the integrated layer of graphene is used as an optical absorber for the materialization of the non-linear activation function. The following Figure 10 presents the application with the use of circuits of free optics for the creation of a series of current peaks with adjustable characteristics of width and breadth [49,82,84,85].

In another survey, the fundamental neuron is based on distributed feedback (DFB) laser of semi-conductors of indium phosphide [86]. The use of this type of laser devices is very common in the construction of SNNs. The laser possesses two photodetectors (PD), which allow for inhibitory as well as excitatory stimuli. The recommended device is very fast, reaching 1012 MACs/sec (MAC—Multiply Accumulate Operations) [87,88].

### 4.6. Reservoir Computing

The use of recurrent neural networks (RNNs) [89,90] has attracted researchers’ interest because of their dynamics. The traditional RNNs, however, present some problems in training and designing, so an evolution has been suggested, namely reservoir computing (RC). It is virtually a neural network of feedback, where the input signals are dependent on time and present maximum efficiency compared to any other architecture in applications of sequence signals such as voice recognition, time series prediction, etc. An RC system consists of a reservoir through which a recording of inputs is conducted in a *n*-dimensional space and a readout layer, where the analysis of standards introduced in the reservoir is performed.

Optical applications with photonic reservoir computing (PRC) architecture are presented in several research projects. In one of them, as shown in Figure 11, passive optical elements are used for the materialization of the reservoir, which consists of a 4 × 4 = 16 node system with splitters, couplers and waveguides, creating in this way a complex interferometer that operates in a random way. The fact that it comprises only passive elements renders it perfect from an energy efficiency point of view, but it displays solely a linear behavior. This can be offset in the readout layer with the introduction of a photodiode as a non-linear element [50,51,91,92].

In Figure 12 is presented a new topology for the reservoir, based on micro-ring resonators (MR), which are non-linear elements and can cover the need for a non-linear transition function, simplifying in this way the readout layer to the fullest extent [51].

The topology in Figure 12, displayed a better error rate compared to others where the reservoir consists of passive linear elements. The reservoir model of our proposed photonic neuron, on the other hand, can change due to collective and synchronous dynamics of the network for spontaneous information processing because the reservoir dynamics can be controlled by tuning optical-pump amplitudes. Network experiments with reservoir neurons revealed that input signals from the correlating neurons can induce an effective change in the pump amplitude. The effective change depends on the increase in the order parameter of synchronization, and it causes spontaneous changes in reservoir modes and firing rates of the networked neurons.

An alternative suggestion for PRC is based on the use of photonic crystal cavity (PCC) in the shape of an ellipse quadrant. This particular architecture proves quite useful for projects of processing optical signals dependent on memory such as the header recognition of digital signals [40,74,91,93].

## 5. Training Methodologies

Training is an important aspect of the neural networks, since it does not only influence the behavior of the network, but also its overall efficiency. In supervised training, the training procedure uses an objectively calculated operation, where the distance (or error) between the desired and the real value is calculated. This operation is used to regulate the internal parameters of the NN, which are the synapses/connections weights of neurons. In order to minimize the deflection between desired and real value, a gradient vector is calculated so that the way of how the error is influenced by any weight shift can be assessed. [49,52,83].

Every time there is a change in the nature of input data in the network, the network needs to be retrained. This retraining can be done gradually as the network performs inference (online learning) [83] or it can be done independently, so that the network can adapt to a new input of training data (offline learning).

Given that the training includes gradient calculation, or even more complex calculations, it is a stage of resources and time consumption. In contrast, the inference (the classification stage by the NN) is a much simpler procedure since the weights are already known in this stage. For this reason, many materializations of PNNs support only the inference stage and the weights are taken with the use of software applications on the level of electronic operation. Moreover, some applications cannot be trained at all, as in [40,42,52,94,95]. These architectures are very fast and efficient as far as energy consumption is concerned, but they are not flexible, as they are especially designed for specific applications as their weights consolidate in the material during their construction.

When in a NN the training is electronic, two main disadvantages appear and, in particular, the physical system dependence on the accuracy of the model is added and the improvement of speed and the efficiency already accomplished with the optical part is eliminated. In order for the training, though complicated, to take full advantage of the photonic technology, it must be specifically adapted to optical architectures.

### 5.1. Propagation

The ONNs offer many advantages as far as the training of NNs is concerned. In a conventional computer, the training is done using the backpropagation error method and the gradient descent application [96]. Nevertheless, in some NNs where the active number of parameters (which are being calculated in every circle) far surpass the number of distinct parameters (as in RNNs and CNNs), the training with backpropagation is definitely ineffective. In particular, the repeated nature of RNNs virtually makes them an extremely deep NN, whereas in CNNs, something relevant happens, since the same weight parameters are used repeatedly in different parts of an image for the output of its characteristics [18,41,97].

For the training of the network with forward propagation, and also for the calculation of gradient in a particular modification step $\Delta_{wij}$ of the weight $w_{ij}$ of a NN, calculations of quantities are needed using the finite difference method (FDM) [98,99,100]:(10)$$f{\left(w_{ij}+\delta_{ij}\right)}_{\;}\mathrm{and}\;f\left(w_{ij}\right)$$

After these two arithmetical operations, we calculate the weight change as follows:(11)$$\Delta w_{ij}=\frac{f\left(w_{ij}+\delta_{ij}\right)-f\left(w_{ij}\right)}{\delta_{ij}}$$

In a conventional computer, the above procedure is computationally costly. On the other hand, in the field of photonic applications, there are suggestions in ONNs that are better at the immediate calculation of the gradient, as every one of the aforementioned steps of propagation is calculated in stable time, which is restricted only by the rate of photo detection, which reaches 100 GHz, and the energy consumption is analogous only to the number of neurons [64].

This particular architecture is capable of reaching performance rates similar or even faster than backpropagation with conventional computers (e.g., in very deep RNNs). Moreover, with the training procedure in the material (on chip), one can easily parameterize and train unitary matrixes, an approach that is particularly useful in deep NNs [72].

Furthermore, in ONNs there is a possibility of training with the backpropagation method, based on the architecture where OIU with MZI are used for the linear operations of multiplications of matrixes [64]. The algorithm of backpropagation training generally operates in a circular mode between two stages, where in the first stage the error propagation is from the end of the network to its beginning, and in the second stage, there is a recalculation of the weights to check the contribution of each one to the output of the network.

In optical materializations, some basic restrictions to the control of weights are present, which musttake into account that $w_{ij}\geq 0$. There cannot be a negative weight value since there is no negative light intensity value [42,49,72]:(12)$$\sum_{i}\;w_{ij}^{\left(l\right)}=1$$

The initial light beam is split into waveguides so that the total of their intensities is stable. These particular restrictions are incorporated with the use of functions, which transform the weights *w* to the desired breadth of activation function values, such as softmax [52,101,102]:(13)$$w_{ij}^{\left(l\right)}=\frac{e^{w_{jj}^{\left(l\right)}}}{\sum_{i}\;e^{w_{ij}^{\left(l\right)}}}$$

In order for backpropagation to be applied, the physical materialization of adjoint variable method (AVM) [103,104] is needed, which allows for the reverse designing of photonic structures. According to this, at first, the adjoint of the initial field is created, the complementary one is propagated in the network reversely to the initial one and the initial field contributes with a replica of the reverse time of the complementary field. After all these, the conditions that yield gradient in every spot are expressed as the solutions of a classical conjugate electromagnetic problem and can be retrieved with an on-the-spot calculation of the field’s intensity. A visualization of the operation of this particular method is presented in Figure 13 [18,41,49,96,97].

This method allows for the effective materialization of backpropagation in a hybrid optical-electronic network, with its main restriction being that a forward feed system, which is mutual and with no losses, is necessary. Moreover, the fact that this method is based on classical Maxwell electromagnetic equations and not on a particular network form renders it extremely flexible for its application on any photonic platform [49,51,70].

### 5.2. Non-Linearity Inversion

In RC photonic applications, the training concerns the readout layer [105,106]. Recently, various researchers on RC have focused their attention on the development of the reservoir with several recommended solutions [107]. Nevertheless, the reading level is of fundamental importance because it ultimately determines the behavior of the network and, unlike the reservoir, must be appropriately trained [51,91]. Hitherto, the training and the conjugation of signals on the reading level has taken place in the conventional electric space, and this resulted in the loss of any gain in speed and energy consumption that the optical part of arrangements introduced [50].

For a fully optical solution in the RC networks, only a simple photo detector is required, which will receive the weighted total of all the optical signals. This approach, however, displays a drawback: we lose the ability for direct observation of the conditions of the photonic reservoir, which is necessary in many linear training algorithms. In order to solve this problem, there is a training procedure presented in Figure 14, in which the reservoir’s states are estimated through a single photodetector at its output, which includes an approximate inversion of the non-linearity of the photodetector, so it was named non-linearity inversion [42,107,108,109].

This method solves the aforementioned issue of direct observation of the reservoir from the reading layer through a PD, with which a calculation of amplitude and its conditions’ phase is materialized. The more complex conditions of the reservoir are observed with the appropriate adaptation of the reading weights, whereas the feedback is achieved through a predetermined input sequence [47,60,89,107].

## 6. Activation Functions

Neurons are the structural element of the network. Each one of these knots receives a total of arithmetic inputs from different sources (either from other neurons or from the environment), performs some computation based on these inputs and produces an output. This output is either directed to the environment or constitutes input to the other neurons of the network. The computational neurons multiply each one of their inputs by the corresponding synaptic weight and calculate the total sum of the products. This total constitutes the activation function definition, which every knot materializes internally. The value that the function takes for this particular definition is also the output of the neuron for the current inputs and weights.

As a result, an important decision that has to be taken into account for the smooth operation of NNs is the selection of the activation function. In bibliographic references, the use of a powerfully non-linear function based on the electro-optical phenomenon is recommended for better results [110]. Respectively, a plethora of non-linear functions have been materialized, which are presented in the next sections [89,107].

### 6.1. z–Transform (Complex Non-Linearity)

This function represents the $Z\;\to \left|Z\right|$ transformation and can be used for full, condense, polar mode. The bilateral z-transform of a sequence of distinguishable time is defined in Equation (14) [111,112,113]:
(14)$$X\left(z\right)=\sum_{n=-\infty }^{+\infty }\;x\left(n\right)\cdot z^{-n}$$
where the complex invariable *z* is called complex frequency and can be expressed with the use of polar coordinates. The *z* transformation of a sequence of distinguishable time is a total of infinite terms, which may converge to a real number for some values of the complex *z* variable and may not converge for some values of the complex *z* variable. The total of the variable values for which the *z* transformation exists, that is, for which the total of *z* transformation converges, constitutes the region of convergence (ROC) [49,114].

The reverse transformation is accomplished by calculating the reverse *z* transformations in each term of the total using *z* transformation pairs and, eventually, using the property of linearity of the *z* transformation. It is materialized with the method of analysis of the rational function in a total of simple fractions as is shown in Equation (15) [17,49]:(15)$$X\left(z\right)=\frac{B\left(z\right)}{A\left(z\right)}=\frac{\sum_{k=0}^{M}\;b\left(k\right)\cdot z^{-k}}{\sum_{k=0}^{N}\;a\left(k\right)\cdot z^{-k}}$$

As an activation function in optical materializations, it is applied in signal analyses and, specifically, in solving linear equations of differences with fixed factors, in calculating the response and in designing linear filters or convolution layers [70].

### 6.2. Electro-Optical Activation (Complex Non-Linearity)

In NN applications of optical components, there is the possibility of creating non-linearity from the already existing material. The activation function is materialized, converting a small part of the power of the input of the optic signal into electrical voltage. The remaining part of the initial optic signal is developed according to phase and amplitude by this voltage as it goes through an interferometer. A typical example of an electro-optical activation function is presented in Figure 15 [60,85,91,109,110]:

For an input signal with a *z* value of amplitude, the non-linear activation function *f(z)* happens as the response of the interferometer and of the components throughout the route of the electric signal as is shown in Equation (16) [15,41,52]:(16)$$f\left(z\right)=j\sqrt{1-a}\cdot \exp\left(-j\left[\frac{g_{\varphi }|z{|}^{2}}{2}+\frac{\varphi_{b}}{2}\right]\right)\cdot \cos\left(\frac{g_{\varphi }|z{|}^{2}}{2}+\frac{\varphi_{b}}{2}\right)z$$
where: $\varphi_{b}=\pi \frac{V_{b}}{V_{\pi }}$ και $g_{\varphi }=\pi \frac{aGRR}{V_{\pi }}$

(1)*α*: the factor of input power transformation into an electric signal.(2)*R*: the response of the photodetector to the optical to electrical unit.(3)*G*: the gain of amplification rate.(4)*V_b_*: the biasing voltage (bias).(5)*V_π_*: the required voltage for the π transformation of the phase.

### 6.3. Sigmoid (Complex Non-Linearity)

The sigmoid activation function is used when a classification between two classes is needed or for a regression of weighted arrangements, as it offers numbers between the space [0, 1] at the output. This can be represented by the transformation shown in Equation (17) [60,77,83]:(17)$$z\to \frac{1}{1+e^{-z}}$$

### 6.4. Softmax (Complex Non-Linearity)

The Softmax function is represented by the transformation [101,102]:(18)$$z\to \frac{e^{z}}{\sum e^{z}}$$

It is mainly used for multiclass problems.

### 6.5. SPM Activation (Non-Linearity)

It represents the transformation [52,110]:(19)$$Z\to Z\cdot e^{\left(-jG|z{|}^{2}\right)}$$
where $G=rad/\left(\frac{V^{2}}{m^{2}}\right)$ is the phase transformation for every unitary change of the input voltage.

### 6.6. zReLU (Non-Linearity)

The *zReLU* is the rectified linear unit function, with which the positive part of its definition is received as follows [10,45,49]:(20)fz=z if Rez>0 and lmz>00

### 6.7. Cosine Activation Function (Non-Linearity)

Many of the recommended optical architectures for NNs use general-purpose equipment (e.g., for optical communications), whereas, ideally, they should be materialized inside a specific material (hardware). Consequently, there is no general approach to the training method of every recommended technique, as each one of them has its own characteristics that should be taken into account. A familiar problem that photonic architectures display concerns the activation function, due to the limited available choices and the difficulty of its materialization. Most of the suggestions use a combination of optical and electronic elements, such as the Mach–Zehnder optical modulators (MZM) [115,116]. The result is a non-linear activation function of cosine form, which is presented in Equation (21) [60,70,76,117]:(21)$$P_{out}=P_{in}\sin^{2}\left(\frac{\pi }{2}\frac{V_{RF}}{V_{\pi }}\right)$$
where $P_{out}$ is the output signal, $P_{in}$ is the continuous wave (CW) under modulation signal, $V_{RF}$ is the input signal of the function and $V_{\pi }$ is the value of input voltage for a phase shift of *π* value.

Another important problem that must be resolved in PNNs is the initialization of their parameters, such as the choice of the initial values of their weights. In their initial definition, the restrictions that exist in every materialization should be taken into consideration, as, for example, the constant bounded response of the signals that go through all the layers of the network. The topology, with which an optical neuron of a cosine activation function is materialized, is shown in Figure 16 [18,49,51,52,117]:

In this particular materialization, two lasers of a different wavelength, *λ_i_(+)* and *λ_i_(−)*, are used, which, through the MZIs functioning as switches (frame *sign of W^(1)^*), are corresponded to positive and negative values of weights, respectively. Afterwards, the signals are led to the modulators (MOD, frame *Input X^(1)^*) so that the input signal can be “printed” on an optical signal of power *P(_Xi_^(1)^)*. The next level (frame *Weight |W^(1)^|*) includes a variable optical attenuator (VOA) [104,105], which is responsible for the amplification of signal-weight as is shown in Relation (22) [18,40,49,117]:(22)$$W_{i}^{\left(1\right)}\cdot P_{x_{i}^{\left(1\right)}}$$

In the next step, the signals are multiplexed (frame *MUX*) and are led in a grade of asynchronous MZI (frame *A-MZI*) for the separation of signals, in signals of positive weight (*λ_1…9(+)_*) and signals of negative weight (*λ_1…9(−)_*), and in the end are added up in photodiodes (blue color). In conclusion, the MZM modulator that follows (MOD) and receives the two signals operates in its non-linear area, materializing the transition function of cosine form. This particular architecture, where each neuron produces a signal that is led to the input of the next neuron, can be completed constructively and constitutes an independent photonic processor (chip) [20,36,84].

## 7. Conclusions

In this research paper, we present an overview of the development and materialization methods of neuromorphic circuits of nanophotonic [61] arrangements for every respective contemporary architecture of conventional neural networks, and the advantages and restrictions that arise during the transition from the electronic to the optical materializations are displayed. The aforementioned networks are energy efficient, when compared to the corresponding electronic ones, and much faster due to photons. The reduction of simultaneous processing time radically increases the potentials of modern computational systems, which use optical arrangements, offering a promising alternative approach to micro-electronic and optical-electronic applications.

All these lead to the conclusion that there are potentials for a full transition to optical materializations as these display the following advantages:(1)Most of the systems do not require energy for the processing of optical signals. As soon as the neural network is trained, the computations on the optical signals are conducted without any additional energy consumption, rendering this particular architecture completely passive.(2)The optical systems, in contrast to the conventional electronic ones, do not produce heat during their operation and, as a result, they can be enclosed in three-dimensional constructions.(3)The processing speed in the optical systems is restricted only by the operation frequency of the laser source of light, which reaches 1 THz.(4)The optical grids enable the multiplication of matrixes with vectors, something which is essential to NNs. The linear transformations (and some non-linear ones) can be performed at the speed of light and detected at a rate of over 100 GHz in photonic networks and, in some cases, with a minimum power consumption.(5)They are not particularly demanding as far as non-linearities are concerned, since many innate optical non-linearities can be used directly for the application of non-linear operations in PNNs, such as the activation functions.

In conclusion, such a system comprises the most efficient, quick and stable circuits of multiple conventional and high non-confining optical technology components for optimal processing, which mimic the key properties of a real brain. 

On the other side, there are some difficulties in the transition to completely PNNs, which are the following:(1)The dimensions of optical devices are analogous to the light wavelength that they use (400 nm–800 nm).(2)The mass production of optical devices is limited compared to the electronic ones, since they lack at least 50 years of research and development.(3)The training of the optical grids is quite difficult because the controlled parameters are active in matrix elements deriving from powerful non-linear functions.(4)The application of matrix transformations with optical components of mass production (such as fibers and lenses) is a restriction to the spread of ONNs due to the need for stability in the signal phase and to the huge number of neurons, which are required in more complex applications.

To summarize, nanophotonics are more expensive and harder to fix, and waveguides and fibers are harder to use than wires and are characterized by spurious reflections that are more troublesome.

Although there are potentials concerning the materialization of PNNs, there are still some areas that require further research, such as some specific architectures of deep neural nets, specifically Long Short-Term Memory Neural Networks, Generative Adversarial Nets, Geometric Deep Neural Networks, Deep Belief Networks and Deep Boltzmann Machines. Due to the significance of DNNs and the role they play in mechanical learning techniques, the research studies should focus on the question whether every type of conventional DNN can be converted in PNN, performing better and, thus, offering more advantages when compared to electronic arrangements. The ultimate goal in this is to replace the huge energy-consuming NNs, with thousands of knots and multiple interconnections among hidden layers, with very fast optical arrangements.

There are also fields where the research on PNNs should focus on, such as the hyper dimensional learning (HL) [118,119], a modern and very promising approach to NNs, which is still in the development stage. Here, the problem of a photonic materialization lies in the very big size of the internal representation of objects that are used in HL.

A further point that needs to be studied is the application of non-linear functions, which in most of the suggestions are materialized through software outside the optical arrangement. This results in the decline of performance, sometimes of a high rate, given that in multilayer NNs it is necessary to insert non-linearity many times successively.

Many more challenges need to be overcome, such as the many different hardware platforms that have been recommended, which are still under investigation with no clear winner yet. Moreover, we have to improve the already developed hardware as, in many cases, basic elements are still simulated, or classic electronic ones are used. Furthermore, a critical element in a recommended NN architecture is its expandability in various applications, something that must be confirmed with further research studies. Finally, the field of NNs, which is still in early stage, is the massive integration of optical arrangements and, of course, their mass production, which is the last and most fundamental fortress of conventional NN arrangements against the transition to fully optical circuits.

## Figures and Tables

**Figure 1 sensors-22-00720-f001:**
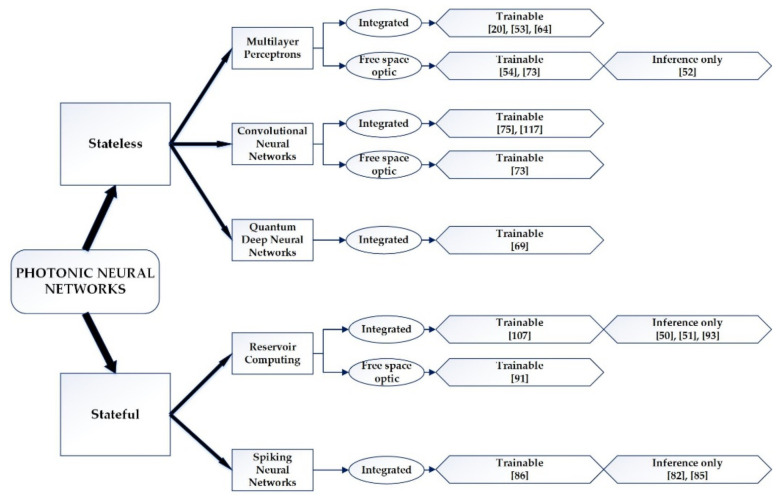
Photonic neural networks classification according to their architecture (stateless or stateful), their design (integrated or free-space optic) and their training ability, presented until 2019.

**Figure 2 sensors-22-00720-f002:**
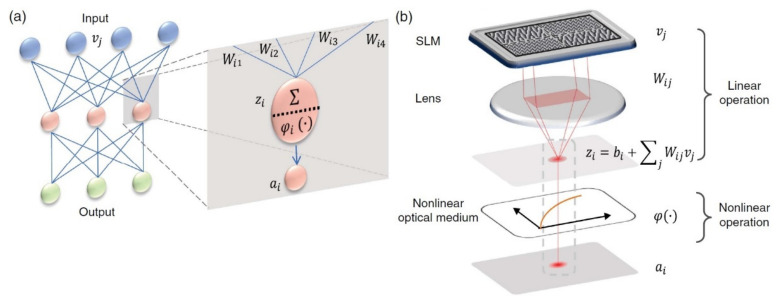
(**a**) A neural network with two layers and a detailed view of one of its neurons. (**b**) Implementation of an optical neuron with linear operation (SLM and lens units) and non-linear operation (activation function *φ*) [54].

**Figure 3 sensors-22-00720-f003:**
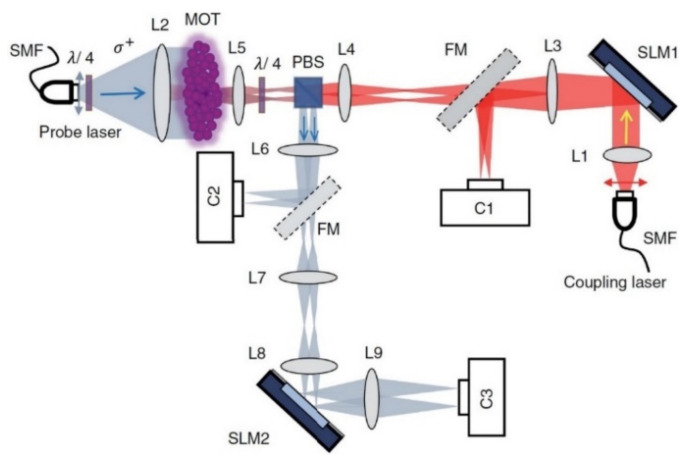
Implementation of the all-optical neural network (AONN) based on free optics [54].

**Figure 4 sensors-22-00720-f004:**
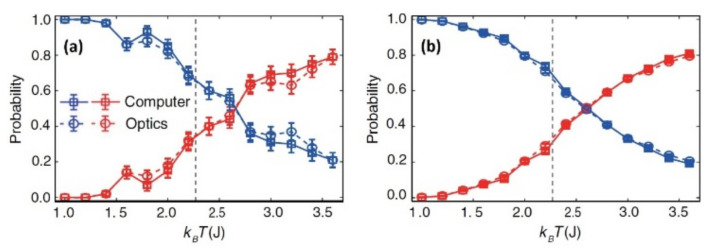
Average possibility of right (blue) and wrong (red) classification of this stage subject to temperature T (K) for 100 (**a**) and 4000 (**b**) settings [54].

**Figure 5 sensors-22-00720-f005:**
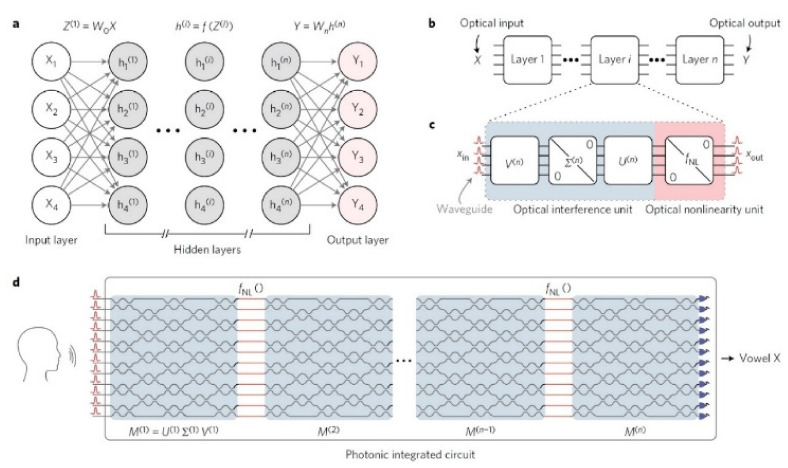
Nanophotonic multilayer perceptron architecture: (**a**) A typical NN with its input–output layers and *n* hidden layers. (**b**) Hidden layers in optical implementation. (**c**) The optical units in each hidden layer. (**d**) The final arrangement in an integrated circuit [64].

**Figure 6 sensors-22-00720-f006:**
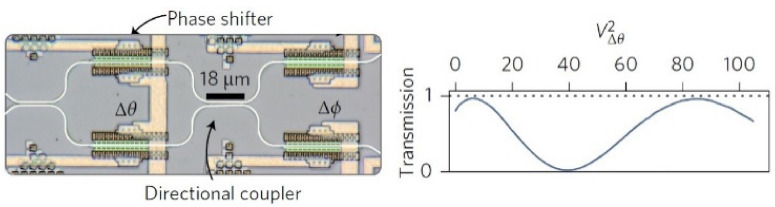
The programmable phase shifter creates modifications in the phase, which, in turn, are converted to amplitude modifications in the directional coupler [64].

**Figure 7 sensors-22-00720-f007:**
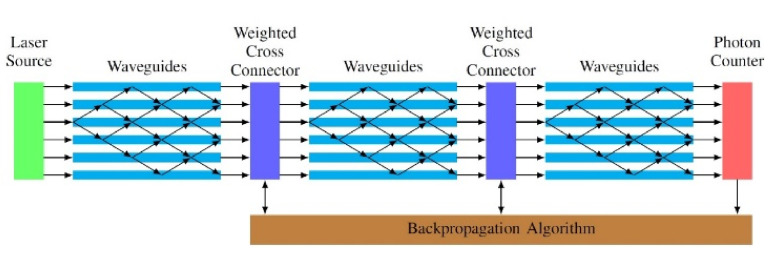
The architecture of a deep photonic neural network (DPNN) [69].

**Figure 8 sensors-22-00720-f008:**
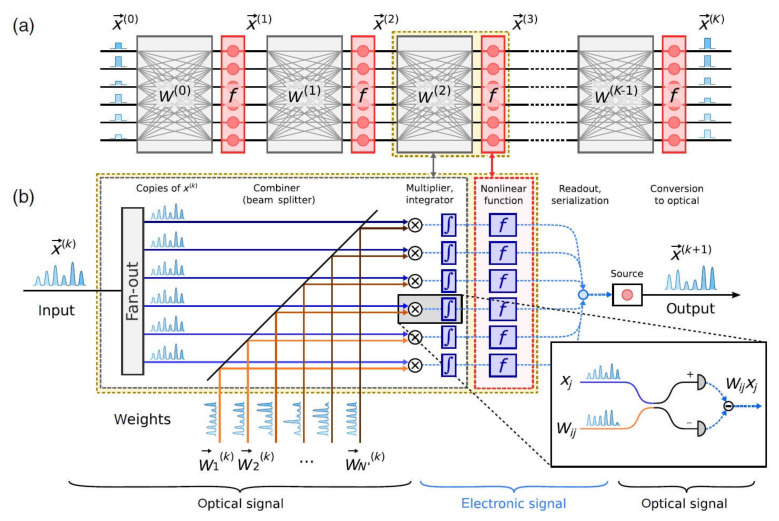
(**a**) Schematic diagram ΝN of K-layers consisting of a multiplier (grey) and an element for the activation function (red). (**b**) The multiplication performs a combination of inputs with the weight signals using homodyning [73].

**Figure 9 sensors-22-00720-f009:**
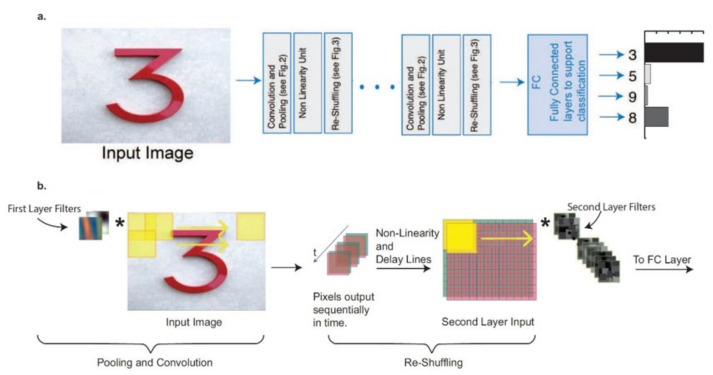
The suggested architecture for a fully optical CNN. (**a**) Logic Block Diagram and (**b**) Schematic Illustration [75].

**Figure 10 sensors-22-00720-f010:**
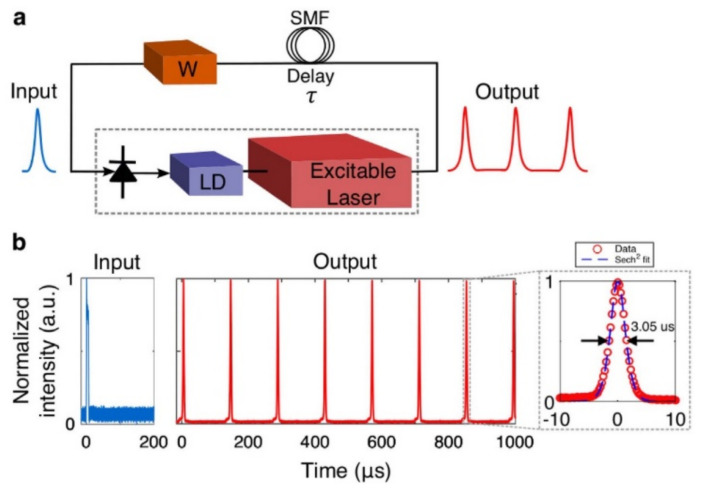
(**a**) The circuit for the creation of repeated current peak. (**b**) The waveforms of the implementation. One pulse of the output is led to the input via single-mode fiber (SMF), which acts as a delay element [82].

**Figure 11 sensors-22-00720-f011:**
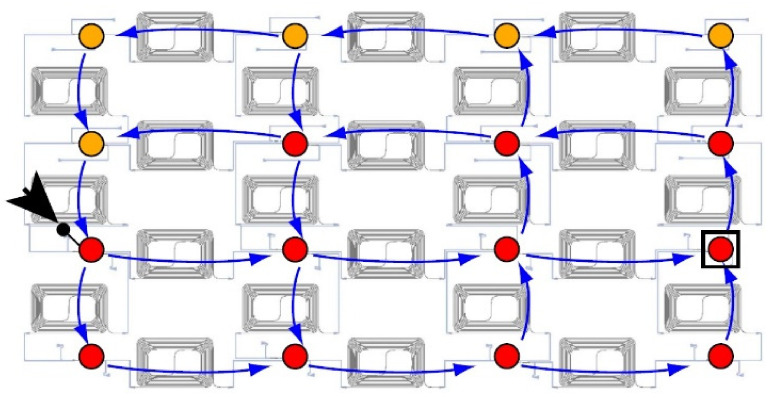
The reservoir structure in optical materialization (chip). It is consisted of interferometers for coupling and splitting between the nodes. Blue arrows represent the specific light flow, if for input is used the node indicated with black arrow. Nodes with yellow dots have output powers below the noise floor. Red ones have an amplitude above noise floor and were measured and used for offline training. For testing the device, an example waveform with sequences of bits with “1” and “0” were collected in the black square with a rounded red dot [50].

**Figure 12 sensors-22-00720-f012:**
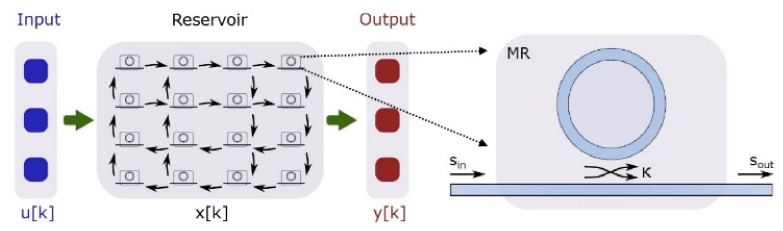
The reservoir with the 16 nodes made from silicon on insulator (SOI) MR [51].

**Figure 13 sensors-22-00720-f013:**
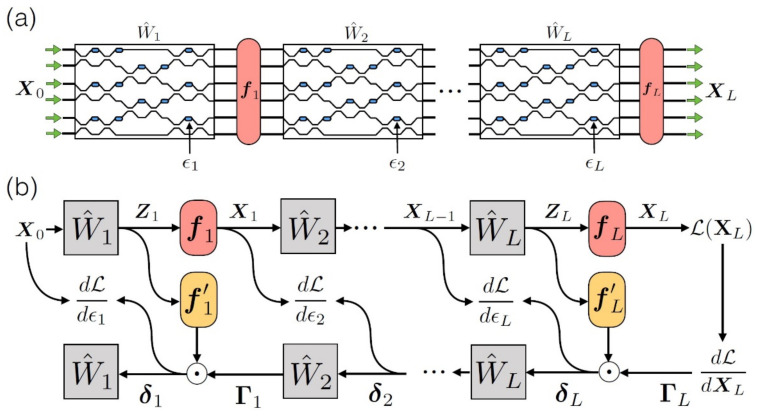
Backpropagation ΡΝΝ. In stage (**a**), the squares correspond to the OIUs, which materialize the linear operation (matrixes $W_{L}$). In blue color, we see the integrated phase shifters for the control of OIU and the training of the network. The red areas correspond to the non-linear activation functions $f_{L}$, which are performed through a computer. Respectively, in stage (**b**), the presentation of the operation for the calculation of NN ranks. The route on top corresponds to the anterior propagation and the bottom to the backpropagation [96].

**Figure 14 sensors-22-00720-f014:**
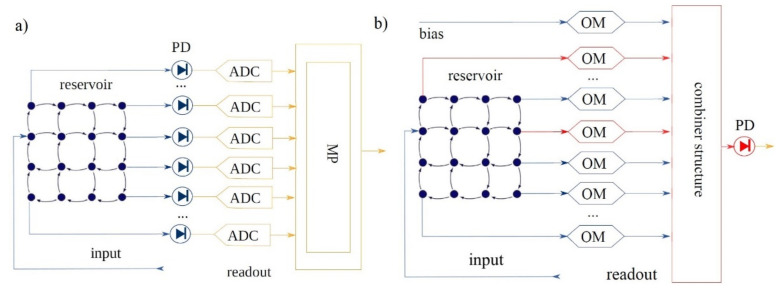
(**a**) The mixed way for training: the optical signal from every node of the reservoir (blue) is transferred through a photodetector (PD) to the electric space (yellow) and through an A/D converter (ADC) to the microprocessor (MP). (**b**) Non-linearity inversion method: the optical signals are modulated (OM) implementing the weights and summed (combiner structure), before converting to electric signal via PD. The states of the reservoir are estimated by setting the weights (red) according to a certain pattern [107].

**Figure 15 sensors-22-00720-f015:**
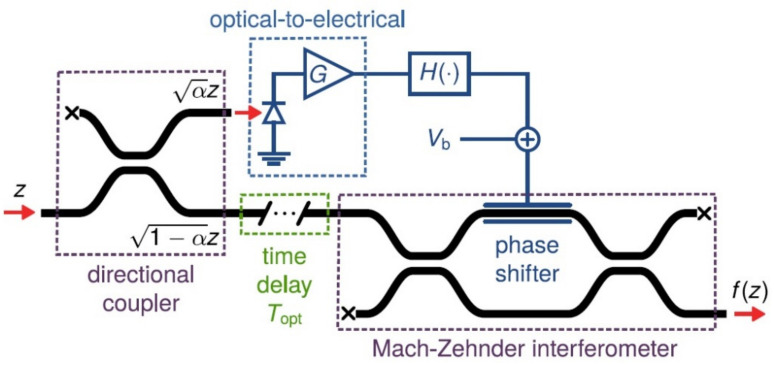
The arrangement for the electro-optical activation function [110].

**Figure 16 sensors-22-00720-f016:**
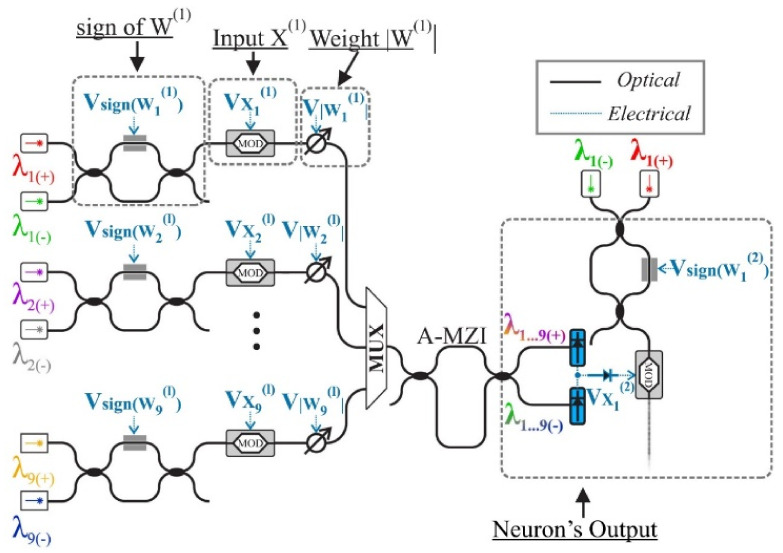
The operation principle of a neuron in an optical materialization [117].

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
