# Peer review of "A Comprehensive Survey on Nanophotonic Neural Networks: Architectures, Training Methods, Optimization, and Activations Functions"

_sensors, 2022, doi:10.3390/s22030720_

Round 1

Reviewer 1 Report

There are no comments. it can be accepted.

Author Response

Thank you for the careful reading and remarks. We deeply appreciate the time and effort you have spent in reviewing our manuscript.

Reviewer 2 Report

Thank you for the corrected version of your manuscript. All my previous comments seem to be addressed appropriately.

Author Response

(The authors gave the same response as above.)

Reviewer 3 Report

All suggestions and comments er taken into account in the current version of the manuscript therefore it can be published.

Author Response

(The authors gave the same response as above.)

Reviewer 4 Report

In the revised manuscript, the miscitation of references is corrected. However, as a review, this paper still has some shortcomings. First, as I pointed out earlier, Figure 1 was copied directly from a review published in 2019. However, photonic neural network is a very new and rapidly developing field, and this graph cannot represent the current development status of this field. The author should redraw the classification tree according to the current literature survey. In addition, at each node of the graph in figure 1, there is a reference number from the original review. Even if the author does not modify the content of the graph, the reference number should be changed to the serial number in the current manuscript. Moreover, I agree with the opinion of the second reviewer. As a review, besides simply listing relevant papers, it should also have its own summary and views, such as comparing the advantages and disadvantages of different methods. Unfortunately, the authors did not supplement this in the revised manuscript. So I still think the paper is not suitable for publication in its current version.

Author Response

Dear respected Reviewer

Thank you for taking the time and effort to review our manuscript. All your questions were answered and the revised changes are highlighted on the manuscript in red color. We are thankful for the comprehensive and valuable suggestions to improve the quality of our work.

  1. In the revised manuscript, the miscitation of references is corrected. However, as a review, this paper still has some shortcomings. First, as I pointed out earlier, Figure 1 was copied directly from a review published in 2019. However, photonic neural network is a very new and rapidly developing field, and this graph cannot represent the current development status of this field. The author should redraw the classification tree according to the current literature survey. In addition, at each node of the graph in figure 1, there is a reference number from the original review. Even if the author does not modify the content of the graph, the reference number should be changed to the serial number in the current manuscript.

A1. Thank you for this constructive comment. A new image was designed, the reports were corrected and new ones were added according to your suggestions.

  1. Moreover, I agree with the opinion of the second reviewer. As a review, besides simply listing relevant papers, it should also have its own summary and views, such as comparing the advantages and disadvantages of different methods. Unfortunately, the authors did not supplement this in the revised manuscript. So I still think the paper is not suitable for publication in its current version.

A2. Thank you for the helpful remarks. This article is an effort to record and study the research that has been conducted concerning the methods of development and materialization of neuromorphic circuits of neural networks of nanophotonic arrangements. In particular, an investigative study of the methods of developing nanophotonic neuromorphic processors, their originality in neuronic architectural structure, their training methods and their optimization has been realized along with the study of specific issues such as optical activation functions and cost functions. Also, according to your suggestions, were added in the Conclusion section an extensive detailed revision of the advantages and disadvantages of nanophotonics. Specifically, “All these lead to the conclusion that there are potentials for a full transition to optical materializations as these display the following advantages:

  1. Most of the systems do not require energy for the processing of optical signals. As soon as the neural network is trained, the computations on the optical signals are conducted without any additional energy consumption, rendering this particular architecture completely passive.
  2. The optical systems, in contrast to the conventional electronic ones, do not produce heat during their operation and, as a result, they can be enclosed in three-dimensional constructions.
  3. The processing speed in the optical systems is restricted only by the operation frequency of the laser source of light, which reaches 1THz.
  4. The optical grids enable the multiplication of matrixes with vectors, something which is essential to NNs. The linear transformations (and some non-linear ones) can be performed at the speed of light and detected at a rate of over 100 GHz in photonic networks and, in some cases, with a minimum power consumption.
  5. They are not particularly demanding as far as non-linearities are concerned, since many innate optical non-linearities can be used directly for the application of non-linear operations in PNNs, such as the activation functions.

In conclusion, such a system comprises the most efficient, quick, and stable circuits, of multiple conventional and high non-confining optical technology components for optimal processing, which mimics key properties of a real brain.

On the other side, there are some difficulties in the transition to completely PNNs, which are the following:

  1. The dimensions of optical devices are analogous to the light wavelength that they use (400nm – 800nm).
  2. The mass production of optical devices is limited compared to the electronic ones, since they lack at least 50 years of research and development.
  3. The training of the optical grids is quite difficult because the controlled parameters are active in matrix elements deriving from powerful non-linear functions.
  4. The application of matrix transformations with optical components of mass production (such as fibers and lenses) is a restriction to the spread of ONNs due to the need for stability in the signal phase and to the huge number of neurons which are required in more complex applications.

To summarize, nanophotonics are more expensive, harder to fix, waveguides and fibers are harder to use than wires and are characterized by spurious reflections that are more troublesome.” 

Reviewer 5 Report

I have reviewed the Manuscript ID: sensors-1541342, having the title "A Comprehensive Survey on Nanophotonic Neural Networks". In this paper, the authors present and analyze the state of art concerning the methods of development and materialization of neuromorphic circuits of Neural Networks of nanophotonic arrangements. I consider that the manuscript will benefit if the authors take into account the following remarks and address within their paper the signaled issues:

Remark 1: the main strong point of the manuscript consists in the fact that it presents an interesting topic for the experts in the field.

Remark 2: the main weak point of the manuscript under review consists in the fact that the authors have submitted their manuscript by specifying the type of their paper as "Article". However, taking into consideration the title and the content of the manuscript, it seems that it is more suitable and has the potential of becoming a "Review" article type rather than an "Article" type that implies bringing novel contributions to the current state of knowledge. The authors should address this inconsistency by deciding what is the type of their paper. In both cases, even if the Manuscript ID: sensors-1541342 is considered an "article" or a "review" paper, the manuscript should be restructured, the authors must perform a thorough editing of the paper in order to improve it. As the authors have mentioned within the title of the manuscript, they aimed to realize a comprehensive survey. The meaning of this concept is a summarization of the evolution of a theory, concept, or technique from inception to the current state of the art. It should focus on collecting and presenting technical information, often to describe the history of discoveries about a given topic. If the authors decide to consider their manuscript as an "article" they should emphasize more their own original contribution to the current state of knowledge, if the authors decide that the manuscript is a "review" one, then the title should identify the manuscript as a systematic review, meta-analysis, or both. In its current form, the title "A Comprehensive Survey on Nanophotonic Neural Networks" lacks specificness, being so broad that hundreds of pages would not be enough to cover the subject properly. Indeed, under the current form the title is more suitable for a book series comprising several volumes. I consider that the title must be adjusted by taking into account the above-mentioned recommendations.

Remark 3: the contribution of the manuscript to the current state of knowledge.       The entire content of the manuscript, all the notions presented, all the equations and all the figures are cited as being retrieved from the scientific literature, while their presentation is rather suitable for a course, an exposition for didactic purposes. The sections' and subsections' titles highlight this aspect: 1. Introduction, 2. Nature of Light, 3. Photonic Neuromorphic Processors, 4. Architectures (comprising the subsections: 4.1 Perceptron, 4.2 Multilayer Perceptrons, 4.3 Deep Photonic Neural Networks, 4.4 Convolutional Neural Networks, 4.5 Spiking Neural Networks, 4.6 Reservoir Computing), 5. Training Methodologies (comprising the subsections: 5.1 Propagation, 5.2 Nonlinearity inversion), 6. Activation Functions (comprising the subsections: 6.1 z–Transform (Complex Non-linearity), 6.2 Electro Optic Activation (Complex Non-linearity),  6.3 Sigmoid (Complex Non-linearity), 6.4 Softmax (Complex Non-linearity),  6.5 SPM Activation (Non-linearity), 6.6 zReLU (Non-linearity), 6.7 Cosine activation function (Non-linearity)) and 7. Conclusions. Several subsections contain only a few lines of text representing a definition (for example subsections 6.3, 6.4, 6.5, 6.6). In the current form of the manuscript, if the intention of the authors was to present a research article, then the elements of originality are lacking. Perhaps the intention of the authors was to present a review article, but the search criteria within the databases, the names of the databases used, the direction that was followed in the presentation are all missing.

Remark 4: the authors have submitted their paper to the Section "Sensor Networks", to the Special Issue "Machine Learning and AI for Sensors" of the MDPI Journal Sensors. In this context, I consider that the authors should strengthen the connection, relationship, and main impact of their study on this domain. In the actual form of the paper, this connection has been explicitly mentioned only once, namely at Line 648: "A modified model of multiple sensors is called multilayer perceptron". It will benefit the paper if the authors provide more details on this issue.

Remark 5: Lines 9-19, the "Abstract" of the paper. The abstract must provide a structured summary including background, objectives, data sources, study eligibility criteria, study appraisal and synthesis methods, results, limitations, conclusions and most important of them all, implications of key findings. I consider that the authors should restructure the abstract as to cover these particularly important points of interest as in the actual form of the manuscript, the abstract offers information related only to some of these aspects and even so, their delimitation is unclear.

Remark 6: Lines 24-179, the "Introduction" section. In the "Introduction" section, the authors did not identify a clear gap in the current state of knowledge that needs to be filled, a gap that is being addressed by their review article and that has not been covered by other existing review articles from the literature. This gap must also be used afterwards, in the final part of the manuscript as well, where the authors should justify why their performed research provides valuable insights that fill the identified gap and provide a possible solution to the problem based on the ensemble of previous studies from the literature that have been reviewed in the manuscript. The authors should highlight what are the main characteristics/findings that make their review study a superior approach when compared to other review studies from the scientific literature that have analyzed similar problems. At the end of the "Introduction" section, the authors must present the structure of their paper under the form: "The rest of the paper is structured as follows: Section 2 contains…".

Remark 7: the rationale and objectives of the study. I consider that the authors did not manage to describe the rationale of their review article in the context of what is already known. The authors should also provide in the introduction an explicit statement of questions that they have devised and addressed with regard to their study design. I consider that the manuscript under review will benefit if the authors explain in the paper what was the criteria in their research methodology based on which they have decided to analyze the chosen papers. In the current form of the manuscript, the methodology does not provide the rationale for the eligibility criteria, a complete rationale for the study design and moreover, the research methodology is not depicted. In will benefit the paper to devise and add to it a series of summarization tables in order to explain the way, the criteria of choosing the papers from which the results presented have been retrieved.

Remark 8: presenting the devised approach. In order to help the readers better understand the methodology of the conducted review, the authors should devise a flowchart that depicts the steps that they have processed in developing their review research and most important of all, the final target. This flowchart will facilitate the understanding of the proposed review approach and at the same time will make the review article more interesting to the readers.

Remark 9: the search queries and the databases. I consider that the methodology lacks an aspect of paramount importance that is mandatory when devising a review article, namely, to present the full electronic search strategy (search queries) for querying the databases, the date and time when the search queries that gathered the initial pool of scientific works were run, including any limits/filters used, such that it could assure the reproducibility of the study. In the current form of the manuscript, the electronic search queries along with the databases used to retrieve the scientific works that have been reviewed have not been stated within the manuscript, therefore affecting the reproducibility of the conducted study. In addition, in the current form of the manuscript, the methodology does not provide the rationale for the eligibility criteria, a complete rationale for the study design.

Remark 10: aspects regarding the type of cited papers. Many references from the authors' study are review articles. I understand perfectly that review articles are a source of good information, but most of the review papers that the authors have cited and used when devising what it should have been their own critical review of the existing state of knowledge, already contain a certain amount of bias, arising from the critical review that the authors of the respective cited papers have performed in their own review articles. Consequently, I consider that the manuscript will benefit if the authors extend the paper by reviewing, in addition to the existing review articles from the manuscript, an increased number of scientific articles that brought contributions to the current state of knowledge from the multiple relevant ones that exist in the scientific literature related to the article's topic, and the authors must review this articles critically from their own perspectives, not from the perspectives of other review articles. There are a lot of valuable studies in the scientific literature that the authors can review, therefore these will highlight even more the personal insights and aspects that the manuscript brings in contrast to other existing scientific works. The authors must review these articles critically from their own perspectives. I consider that the literature review should be improved by performing a careful analysis of the cited works. The authors must highlight for the involved referenced papers the main contribution that the authors of the referenced papers have brought to the current state of knowledge, the methods used by the authors of the referenced papers, a brief presentation of the main obtained results and some limitations of the referenced article. This is the only way to contextualize the current state of the art in which the authors of the manuscript position their paper, identify and address aspects that have not been tackled/solved yet by the existing studies.

Remark 11: discussing the obtained results. A point of interest that I consider the authors must analyze consists in pointing out the added benefit that the new insights obtained after performing their research brings to the current state of knowledge. The authors should strengthen what added knowledge will the experts in the field have after having read the results obtained so that these will help them in their activity. The paper will benefit if the authors make a step further, beyond their approach and provide an insight at the end of the paper regarding what they consider to be, based on the obtained results, the most important steps that all the involved parties should take in order to benefit from the results obtained after the authors have performed the review within the manuscript. The authors should also highlight current limitations of their study, and briefly mention some precise directions that they intend to follow in their future research work.

Minor remark:

  • Lines 1870-1871: "Long-Short-Term Memory Neural Networks, Generative Adversarial Nets, Geometric Deep Neural Networks, Deep Belief Networks, Deep Boltzmann Machines, etc." In a scientific paper one should avoid using run-on expressions, such as "and so forth", "and so on" or "etc.". Therefore, instead of "etc.", the sentence should mention all the elements that are relevant to the manuscript.

Author Response

Dear respected Reviewer

Thank you for taking the time and effort to review our manuscript. All your questions were answered and the revised changes are highlighted on the manuscript in red color. We are thankful for the comprehensive and valuable suggestions to improve the quality of our work.

I have reviewed the Manuscript ID: sensors-1541342, having the title "A Comprehensive Survey on Nanophotonic Neural Networks". In this paper, the authors present and analyze the state of art concerning the methods of development and materialization of neuromorphic circuits of Neural Networks of nanophotonic arrangements. I consider that the manuscript will benefit if the authors take into account the following remarks and address within their paper the signaled issues:

  1. Remark 1: the main strong point of the manuscript consists in the fact that it presents an interesting topic for the experts in the field.

We are grateful for your consideration of this manuscript

  1. Remark 2: the main weak point of the manuscript under review consists in the fact that the authors have submitted their manuscript by specifying the type of their paper as "Article". However, taking into consideration the title and the content of the manuscript, it seems that it is more suitable and has the potential of becoming a "Review" article type rather than an "Article" type that implies bringing novel contributions to the current state of knowledge. The authors should address this inconsistency by deciding what is the type of their paper. In both cases, even if the Manuscript ID: sensors-1541342 is considered an "article" or a "review" paper, the manuscript should be restructured, the authors must perform a thorough editing of the paper in order to improve it. As the authors have mentioned within the title of the manuscript, they aimed to realize a comprehensive survey. The meaning of this concept is a summarization of the evolution of a theory, concept, or technique from inception to the current state of the art. It should focus on collecting and presenting technical information, often to describe the history of discoveries about a given topic. If the authors decide to consider their manuscript as an "article" they should emphasize more their own original contribution to the current state of knowledge, if the authors decide that the manuscript is a "review" one, then the title should identify the manuscript as a systematic review, meta-analysis, or both. In its current form, the title "A Comprehensive Survey on Nanophotonic Neural Networks" lacks specificness, being so broad that hundreds of pages would not be enough to cover the subject properly. Indeed, under the current form the title is more suitable for a book series comprising several volumes. I consider that the title must be adjusted by taking into account the above-mentioned recommendations.

Thank you for the helpful remarks. Must be mentioned that the type of the paper as "Article" was decided after review by the editorial team of the MDPI journal from the preliminary checks/corrections. This article is an effort to record and study the research that has been conducted concerning the methods of development and materialization of neuromorphic circuits of neural networks of nanophotonic arrangements. In particular, an investigative study of the methods of developing nanophotonic neuromorphic processors, their originality in neuronic architectural structure, their training methods and their optimization has been realized along with the study of specific issues such as optical activation functions and cost functions. In this point of view, the new title is "A Comprehensive Survey on Nanophotonic Neural Networks: Architectures, Training Methods, Optimization, and Activations Functions".

  1. Remark 3: the contribution of the manuscript to the current state of knowledge. The entire content of the manuscript, all the notions presented, all the equations and all the figures are cited as being retrieved from the scientific literature, while their presentation is rather suitable for a course, an exposition for didactic purposes. The sections' and subsections' titles highlight this aspect: 1. Introduction, 2. Nature of Light, 3. Photonic Neuromorphic Processors, 4. Architectures (comprising the subsections: 4.1 Perceptron, 4.2 Multilayer Perceptrons, 4.3 Deep Photonic Neural Networks, 4.4 Convolutional Neural Networks, 4.5 Spiking Neural Networks, 4.6 Reservoir Computing), 5. Training Methodologies (comprising the subsections: 5.1 Propagation, 5.2 Nonlinearity inversion), 6. Activation Functions (comprising the subsections: 6.1 z–Transform (Complex Non-linearity), 6.2 Electro Optic Activation (Complex Non-linearity),  3 Sigmoid (Complex Non-linearity), 6.4 Softmax (Complex Non-linearity),  6.5 SPM Activation (Non-linearity), 6.6 zReLU (Non-linearity), 6.7 Cosine activation function (Non-linearity)) and 7. Conclusions. Several subsections contain only a few lines of text representing a definition (for example subsections 6.3, 6.4, 6.5, 6.6). In the current form of the manuscript, if the intention of the authors was to present a research article, then the elements of originality are lacking. Perhaps the intention of the authors was to present a review article, but the search criteria within the databases, the names of the databases used, the direction that was followed in the presentation are all missing.

Thank you for your careful reading. The main contribution of this research work is that is the first time in the literature that was presented in a single paper the most well-known architectures, training methods, optimization, and activations functions of the nanophotonins networks. Also, according to your suggestions, were added in the Conclusion section an extensive detailed meta-review analysis of the advantages and disadvantages of nanophotonics networks. Specifically, “All these lead to the conclusion that there are potentials for a full transition to optical materializations as these display the following advantages:

  1. Most of the systems do not require energy for the processing of optical signals. As soon as the neural network is trained, the computations on the optical signals are conducted without any additional energy consumption, rendering this particular architecture completely passive.
  2. The optical systems, in contrast to the conventional electronic ones, do not produce heat during their operation and, as a result, they can be enclosed in three-dimensional constructions.
  3. The processing speed in the optical systems is restricted only by the operation frequency of the laser source of light, which reaches 1THz.
  4. The optical grids enable the multiplication of matrixes with vectors, something which is essential to NNs. The linear transformations (and some non-linear ones) can be performed at the speed of light and detected at a rate of over 100 GHz in photonic networks and, in some cases, with a minimum power consumption.
  5. They are not particularly demanding as far as non-linearities are concerned, since many innate optical non-linearities can be used directly for the application of non-linear operations in PNNs, such as the activation functions.

In conclusion, such a system comprises the most efficient, quick, and stable circuits, of multiple conventional and high non-confining optical technology components for optimal processing, which mimics key properties of a real brain.

On the other side, there are some difficulties in the transition to completely PNNs, which are the following:

  1. The dimensions of optical devices are analogous to the light wavelength that they use (400nm – 800nm).
  2. The mass production of optical devices is limited compared to the electronic ones, since they lack at least 50 years of research and development.
  3. The training of the optical grids is quite difficult because the controlled parameters are active in matrix elements deriving from powerful non-linear functions.
  4. The application of matrix transformations with optical components of mass production (such as fibers and lenses) is a restriction to the spread of ONNs due to the need for stability in the signal phase and to the huge number of neurons which are required in more complex applications.

To summarize, nanophotonics are more expensive, harder to fix, waveguides and fibers are harder to use than wires and are characterized by spurious reflections that are more troublesome.”

  1. Remark 4: the authors have submitted their paper to the Section "Sensor Networks", to the Special Issue "Machine Learning and AI for Sensors" of the MDPI Journal Sensors. In this context, I consider that the authors should strengthen the connection, relationship, and main impact of their study on this domain. In the actual form of the paper, this connection has been explicitly mentioned only once, namely at Line 648: "A modified model of multiple sensors is called multilayer perceptron". It will benefit the paper if the authors provide more details on this issue.

We would like to thank the reviewer for this constructive comment that gives us the chance to clarify things further.  All the presented applications and standardizations in this paper are systems-on-chip (sensors). We quote the most relevant part that explains this point "The biggest challenge in this field is the development of fully functional and utilizable neuromorphic systems-on-chip (NSoC) [10], [13], which will be able to approach the biological human intelligence, performing the same tasks that the human brain effortlessly performs in no time at all and without remarkable consumption of resources and energy [14]. The neuromorphic computation comprises the creation of neural networks in the matter, where the neurons of a physical device are connected with the corresponding synapses of physical devices [15]. The main motive for the neuromorphic computation is the time needed to process the computations and the energy performance provided by a distributed architecture that avoids the energy turbulence of data between the memory and the CPU [16].

The NSoCs based solidly on previous computational technology overcome the von Neumann congestion, massively use simultaneous computational procedures, and are tolerant to faults [17]. Essentially, they form the way in which neural networks function conveying information in the same temporal and spatial way as the human brain. Furthermore, taking advantage of techniques such as the memristors [10], they are capable of modelizing learning skills; that is the adjustment ability of synapses in storing and conveying information depending on the evolution of a dynamic situation [16].

However, the most important development in the application efforts and standardization of NSoCs is spotted in the expanded efforts for developing fully optical neural net-works (ONNs), also known as photonic neural networks (PNNs) or nanophotonic neural networks (NNNs) [15], [18]–[21]. The previously mentioned systems are based on the evolutions of optical technology and the most recent researches concerning photonics. Photonics is the science and technology field that deals with the creation, control, and detection of photons, especially in the area of visible light and the near-infrared electromagnetic spectrum (wavelength, polarization, transmission rate etc.) and the great potentials of their inter-connection [20], [22], [23]. It is directly related, in basic as well as applied research, to quantum optics as to how linear transformations can be applied with the minimum energy consumption and with the slightest latency time on neuron level and with optoelectronics in the study of active and passive materials which interact electrically with light [24], [25]".

  1. Remark 5: Lines 9-19, the "Abstract" of the paper. The abstract must provide a structured summary including background, objectives, data sources, study eligibility criteria, study appraisal and synthesis methods, results, limitations, conclusions and most important of them all, implications of key findings. I consider that the authors should restructure the abstract as to cover these particularly important points of interest as in the actual form of the manuscript, the abstract offers information related only to some of these aspects and even so, their delimitation is unclear.

The following added in the abstract “The main contribution of this research work is that is the first time in the literature that was presented in a single paper the most well-known architectures, training methods, optimization, and activations functions of the nanophotonins networks. Also including an extensive detailed meta-review analysis of the advantages and disadvantages of nanophotonics networks.” So, the revised abstract offers detailed information related to the motivation of the paper aspects and now is clear the contribution and the aim of key findings. Thank you for this comment.

  1. Remark 6: Lines 24-179, the "Introduction" section. In the "Introduction" section, the authors did not identify a clear gap in the current state of knowledge that needs to be filled, a gap that is being addressed by their review article and that has not been covered by other existing review articles from the literature. This gap must also be used afterwards, in the final part of the manuscript as well, where the authors should justify why their performed research provides valuable insights that fill the identified gap and provide a possible solution to the problem based on the ensemble of previous studies from the literature that have been reviewed in the manuscript. The authors should highlight what are the main characteristics/findings that make their review study a superior approach when compared to other review studies from the scientific literature that have analyzed similar problems. At the end of the "Introduction" section, the authors must present the structure of their paper under the form: "The rest of the paper is structured as follows: Section 2 contains…".

Thank you for the remarks. In the revised version of the paper mentioned and presented clearly the purpose of this work and the gap that it fills. Also, the structure of the paper is added in the end of the introduction section. Specifically, “Many efforts have been made in recent years to shift from conventional electronics to optical circuits. This review records the most recent research to clarify how close we are to the complete transition to photonic arrangements and their exceptional prospects. More-over, the main contribution of this research work is that is the first time in the literature that was presented in a single paper the most well-known architectures, training methods, optimization, and activations functions of the nanophotonins networks. Also, the manu-script includes an extensive detailed meta-review analysis of the advantages and disad-vantages of nanophotonics networks.

The rest of the paper is organized as follows: Section 2 contains the principles of light and matter interaction. Section 3 describes the current state of research in neuromorphic processors using photon circuits and section 4 analyzes these architectures. Section 5 is dedicated to the training operation of PNNs. Section 6 summarizes the most common activation functions which are used. In conclusion, section 7 presents the final remarks and perspectives.”

  1. Remark 7: the rationale and objectives of the study. I consider that the authors did not manage to describe the rationale of their review article in the context of what is already known. The authors should also provide in the introduction an explicit statement of questions that they have devised and addressed with regard to their study design. I consider that the manuscript under review will benefit if the authors explain in the paper what was the criteria in their research methodology based on which they have decided to analyze the chosen papers. In the current form of the manuscript, the methodology does not provide the rationale for the eligibility criteria, a complete rationale for the study design and moreover, the research methodology is not depicted. In will benefit the paper to devise and add to it a series of summarization tables in order to explain the way, the criteria of choosing the papers from which the results presented have been retrieved.

In the revised paper mentioned the purpose and the motivation of this work and the gap that it fills, as mentioned above. Thank you for this helpful comment.

  1. Remark 8: presenting the devised approach. In order to help the readers better understand the methodology of the conducted review, the authors should devise a flowchart that depicts the steps that they have processed in developing their review research and most important of all, the final target. This flowchart will facilitate the understanding of the proposed review approach and at the same time will make the review article more interesting to the readers.

Thank you for this constructive remark. Figure 1, depicted a flowchart of the photonic neural networks classification according to their architecture (stateless or stateful), their design (integrated or free space optic), and their training ability. In addition, this flowchart includes a classification tree according to the current literature survey. For each node of the graph, there is a reference number from the current manuscript.

  1. Remark 9: the search queries and the databases. I consider that the methodology lacks an aspect of paramount importance that is mandatory when devising a review article, namely, to present the full electronic search strategy (search queries) for querying the databases, the date and time when the search queries that gathered the initial pool of scientific works were run, including any limits/filters used, such that it could assure the reproducibility of the study. In the current form of the manuscript, the electronic search queries along with the databases used to retrieve the scientific works that have been reviewed have not been stated within the manuscript, therefore affecting the reproducibility of the conducted study. In addition, in the current form of the manuscript, the methodology does not provide the rationale for the eligibility criteria, a complete rationale for the study design.

This work is a part of long-term research of photonics and the main innovations in this field. It is not a part of an electronic search strategy in the most relevant databases. Unfortunately, the date and time when the search queries that were gathered is not can be tracked, because the initial pool of this scientific work started 4 years prior. Of course, there are all the appropriate references that can contribute to the reproducibility of the conducted study. Thank you for this comment.

  1. Remark 10: aspects regarding the type of cited papers. Many references from the authors' study are review articles. I understand perfectly that review articles are a source of good information, but most of the review papers that the authors have cited and used when devising what it should have been their own critical review of the existing state of knowledge, already contain a certain amount of bias, arising from the critical review that the authors of the respective cited papers have performed in their own review articles. Consequently, I consider that the manuscript will benefit if the authors extend the paper by reviewing, in addition to the existing review articles from the manuscript, an increased number of scientific articles that brought contributions to the current state of knowledge from the multiple relevant ones that exist in the scientific literature related to the article's topic, and the authors must review this articles critically from their own perspectives, not from the perspectives of other review articles. There are a lot of valuable studies in the scientific literature that the authors can review, therefore these will highlight even more the personal insights and aspects that the manuscript brings in contrast to other existing scientific works. The authors must review these articles critically from their own perspectives. I consider that the literature review should be improved by performing a careful analysis of the cited works. The authors must highlight for the involved referenced papers the main contribution that the authors of the referenced papers have brought to the current state of knowledge, the methods used by the authors of the referenced papers, a brief presentation of the main obtained results and some limitations of the referenced article. This is the only way to contextualize the current state of the art in which the authors of the manuscript position their paper, identify and address aspects that have not been tackled/solved yet by the existing studies.

Thank you for your remarks. There are 119 references at work, of which about 25% are review papers. Consequently, we consider that the manuscript reviewed an important number of scientific articles that brought contributions to the current state of knowledge from the relevant ones that exist in the scientific literature related to the article's topic.  On the other hand, there is an important disadvantage that restricts the review process, the insights, and aspects relevant to a nanophotonics neural network manuscript: most of them cannot reproducible. So, the review process in order to be fair must be related to other reviews studies.  In any case, the presented work highlights for the involved referenced papers the main contribution that the authors of the referenced papers have brought to the current state of knowledge. An extensive detailed meta-review analysis of the advantages and disadvantages of cited papers there are in the conclusion section.

  1. Remark 11: discussing the obtained results. A point of interest that I consider the authors must analyze consists in pointing out the added benefit that the new insights obtained after performing their research brings to the current state of knowledge. The authors should strengthen what added knowledge will the experts in the field have after having read the results obtained so that these will help them in their activity. The paper will benefit if the authors make a step further, beyond their approach and provide an insight at the end of the paper regarding what they consider to be, based on the obtained results, the most important steps that all the involved parties should take in order to benefit from the results obtained after the authors have performed the review within the manuscript. The authors should also highlight current limitations of their study, and briefly mention some precise directions that they intend to follow in their future research work.

Thank you for the remarks. The conclusion section was revised according to your suggestions: “Although there are potentials concerning the materialization of PNNs, there are still some areas which require further research, such as some specific architectures of deep neural nets and specifically Long-Short-Term Memory Neural Networks, Generative Adversarial Nets, Geometric Deep Neural Networks, Deep Belief Networks and Deep Boltzmann Machines. Due to the significance of DNNs and the role they play in mechanical learning techniques, the research studies should focus on the question whether every type of conventional DNN can be converted in PNN, performing better and, thus, offering more advantages when compared to electronic arrangements. The ultimate goal in this is to replace the huge energy consuming NNs, with thousands of knots and multiple interconnections among hidden layers, with very fast optical arrangements.

There are also fields where the research on PNNs should focus on, such as the hyper dimensional learning (HL) [118], [119], a modern and very promising approach to NNs, which is still in the development stage. Here, the problem of a photonic materialization lies in the very big size of the internal representation of objects that are used in HL.

A further point that needs to be studied is the application of non-linear functions, which in most of the suggestions are materialized through software outside the optical arrangement. This results in the decline of performance, sometimes of a high rate, given that in multilayer NNs it is necessary to insert non-linearity many times successively.

Many more challenges need to be overcome, such as the many different hardware platforms that have been recommended, which are still under investigation with no clear winner yet. Moreover, we have to improve the already developed hardware as, in many cases, basic elements are still simulated or classic electronic ones are used. Furthermore, a critical element in a recommended NN architecture is its expandability in various applications, something which must be confirmed with further research studies. Finally, the field of NNs, which is still in early stage, is the massive integration of optical arrangements and, of course, their mass production, which is the last and most fundamental for-tress of conventional NN arrangements against the transition to fully optical circuits.”

  1. Minor remark: Lines 1870-1871: "Long-Short-Term Memory Neural Networks, Generative Adversarial Nets, Geometric Deep Neural Networks, Deep Belief Networks, Deep Boltzmann Machines, etc." In a scientific paper one should avoid using run-on expressions, such as "and so forth", "and so on" or "etc.". Therefore, instead of "etc.", the sentence should mention all the elements that are relevant to the manuscript.

Thank you for the careful reading and suggestions. The sentence was revised.

Round 2

Reviewer 4 Report

The paper has been improved following all the suggestions provided by the reviewers. Therefore, I suggest the publication of the revised manuscript. 

Reviewer 5 Report

I have reviewed the revised version of the Manuscript ID: sensors-1541342, having the revised title "A Comprehensive Survey on Nanophotonic Neural Networks: Architectures, Training Methods, Optimization, and Activations Functions" and I can state that the manuscript has been improved in contrast to the previous submission.

This manuscript is a resubmission of an earlier submission. The following is a list of the peer review reports and author responses from that submission.

Round 1

Reviewer 1 Report

This manuscript is a review of nanophotonic neural networks. This is very an interesting topic. However, the authors was very perfunctory when writing this review, which contained many errors and inaccuracies. Figure 1 is from ref. [49], another review in 2019, and the authors did not even modify the reference number in the figure when coping it. On page 7, references [27] and [29] are cited to illustrate Figure 3. However, these two papers are about laser machining and are not related to the content. Figures 5 and 6 were originally from the paper (Nature Photonics, 2017, 11(7): 441-446). However, the author did not cite this paper but cited some irrelevant papers. Similarly, Fig. 8 is from the paper (Physical Review X, 2019, 9(2): 021032), and fig. 13 is from the paper (Optica, 2018, 5(7): 864-871). None of these have been correctly cited. There are many references cited in the article that are not consistent with the content. In addition, the manuscrpt also contains a lot of inaccurate content, and almost no discussion of nanophotonics. Therefore I think this manuscript should be rejected.

Reviewer 2 Report

In this paper, the authors aim to record and study the research that has been conducted concerning the methods of development and materialization of neuromorphic circuits of Neural Networks of nanophotonic arrangements. In general, the reviewer finds that this paper just summarizes published papers. Therefore, the novelty in the paper is plain. The authors should further highlight their novelty.

  1. The authors just summarize various papers. Many figures are directly cited from references. Therefore, the reviewer wanders to know what the authors’ work is.

  1. In this paper, the reviewer cannot find the comparison among various papers. The authors should highlight their own work rather than just summarizing published papers. For example, the authors can compare various methods. The relationships, advantages and disadvantages related to various papers should be further clarified. A table is usually suitable for this comparison.

Reviewer 3 Report

The presented paper is a very exhaustive and complete survey on nanophotonics neural networks. The list of sources shows the big effort the authors put into the preparation of this study. Most of them are less than 10 years old and thus contain contemporary and novel knowledge. 

The authors described all technological issues and methods, including photonic neuromorphic processors, architectures, training methodologies, and activation functions. 

In my opinion, to make this study full there is a need for some examples, practical solutions, and comparisons.

Reviewer 4 Report

Please find the detailed review in the attached PDF file. 
